# A Cell for the Ages: Human γδ T Cells across the Lifespan

**DOI:** 10.3390/ijms21238903

**Published:** 2020-11-24

**Authors:** Brandi L. Clark, Paul G. Thomas

**Affiliations:** 1Department of Immunology, St. Jude Children’s Research Hospital, Memphis, TN 38105, USA; brandi.clark@stjude.org; 2Integrated Biomedical Sciences Program, University of Tennessee Health Science Center, Memphis, TN 38163, USA

**Keywords:** γδ T cells, human, development, age, periphery, passive immunity, reactive immunity, cancer, transplant, cytomegalovirus, influenza, tuberculosis

## Abstract

The complexity of the human immune system is exacerbated by age-related changes to immune cell functionality. Many of these age-related effects remain undescribed or driven by mechanisms that are poorly understood. γδ T cells, while considered an adaptive subset based on immunological ontogeny, retain both innate-like and adaptive-like characteristics. This T cell population is small but mighty, and has been implicated in both homeostatic and disease-induced immunity within tissues and throughout the periphery. In this review, we outline what is known about the effect of age on human peripheral γδ T cells, and call attention to areas of the field where further research is needed.

## 1. Introduction

Age and functionality of the immune system are intricately connected, as has been well established throughout the literature in the fields of both innate and adaptive immunology [1,2,3,4,5,6,7,8,9]. These differences are often characterized based on three main phases of life: infancy, adulthood, and old age. Interestingly, there tends to be overlap in the immune function on opposite ends of the spectrum—namely that children and the elderly often respond to immune challenges with similar results. That said, the mechanisms behind these responses are quite different in nature. An example of this can be seen when one compares the innate immune responses triggered by Toll-like receptor (TLR) stimulation of infants and the elderly [10]. Both populations are highly susceptible to infection and increased inflammation, yet the underlying mechanisms of these phenomena are unique. Elderly individuals suffer from inflamm-aging, a condition where baseline inflammatory levels are heightened in aging individuals. However, their TLR sensing capabilities are often decreased, preventing a robust inflammatory response at the proper time. In infants, TLR sensors seem to be functioning properly—the difference is that the downstream responses are unique compared to traditional adult responses, including a decrease in TLR-mediated innate effector molecules, an increased susceptibility to oxygen radicals, and differing cytokine production [10].

When considering the effects of aging on the immune system, there are many other factors that should be included—such as environmental conditions and prior antigen exposure [11,12]. Lifestyle choices also contribute to immune aging, as nutrition, pharmacologic intervention, and psychological stress have all been shown to affect the immune system [13,14]. For these reasons, chronological age alone is not the most robust correlate for immune aging. To combat these discrepancies, a study published in 2019 outlined a new metric for more accurately determining immune system age, coined “IMM-AGE” [15]. This high-dimensional trajectory algorithm allows for a more precise aging of the immune system on an individual basis.

Even as our understanding of the effects of age on the immune system is expanding, there are still many gaps in the field. A notable example of this can be observed in γδ T cells. This cell subset of the adaptive immune system uniquely exhibits characteristics of both innate and adaptive immunity, and is often thought to be “bridging the gap” between the two arms of the immune system [16,17]. Age has been associated with functional change in γδ T cells in mice and humans [18], but in most studies these effects are too often overlooked, especially in human research. In this review, we outline the reported role that age plays in the human γδ T cell response in health and disease, as well as highlighting areas of the field that are sorely lacking an age component.

## 2. Passive γδ T Cell Immunity

The immune system broadly consists of two arms: innate and adaptive immunity. The innate immune system is considered the first line of defense against pathogens and damage, rapidly responding in a nonspecific manner. The adaptive immune system is slower to respond, and elicits its functions in a pathogen-specific manner. T cells and B cells make up the adaptive compartment and achieve their specificity via their T cell receptor (TCR) and B cell receptor (BCR), respectively. The TCR and BCR are both created by V(D)J Recombination, a semi-random process by which lymphocytes construct their antigen-specific receptors [19]. The TCR is composed of either an alpha and beta chain or a gamma and delta chain, and the combination of all TCRs in an individual is referred to as their TCR repertoire. Analysis of the TCR repertoire can provide insights into antigen reactivity and specificity, as well as history of pathogen exposure [20]. αβ T cells express CD8 or CD4, and are restricted to recognizing antigenic peptides presented by a Major Histocompatibility Complex (MHC) in a CD3-dependent manner. γδ T cells were first identified and characterized in the 1980s [21,22,23], and they are similar to αβ T cells in many regards. This cell type has a TCR that is also constructed via V(D)J Recombination, and utilizes CD3 to respond in an antigen-specific manner. γδ T cells are much less frequent than αβ T cells, making up only 5–10% of CD3+ cells in peripheral blood, although they are present at higher frequencies in mucosal tissues. They are MHC-unrestricted, meaning that their recognition of pathogens does not require MHC presentation. Additionally, while αβ T cells are only able to recognize antigenic peptide fragments, γδ T cells can also recognize entire proteins and stress signals. This recognition occurs via Natural Killer (NK) cell receptors and Pattern Recognition Receptors (PRRs), in addition to their TCR. The myriad of receptors expressed by γδ T cells allows this population to respond in an innate-like manner to pathogen- and damage-associated molecular patterns (PAMPs and DAMPs) [17,24].

Human γδ T cells have seven functional T-cell receptor gamma variable region (*TRGV*)-encoding genes and eight functional T-cell receptor delta variable region (*TRDV*)-encoding genes [25]. Broadly speaking, these cells are often split into two major groups based on *TRDV* usage: Vδ2+ and Vδ2− T cells [26,27]. The majority of Vδ2+ T cells are Vγ9+ (Lefranc nomenclature [28], formerly known as Vγ2+), and make up over 90% of γδ T cells in peripheral blood [29]. Traditionally regarded as the innate-like subset, Vγ9+Vδ2+ respond to both endogenous and exogenous phosphoantigens (pAgs), such as (E)-4-hydroxy-3-methyl-but-2-enyl pyrophosphate (HMB-PP) and Isopentenyl pyrophosphate (IPP) [25]. They utilize a semi-invariant, oligoclonal repertoire [29], and their rapid BTN3A1/BTN2A1-dependent functionality occurs even in the absence of previous exposure [30,31,32,33]. Vδ2− γδ T cells are largely Vδ1+, although Vδ3+ and Vδ5+ segments are also observed. Unlike Vγ9+Vδ2+ T cells, Vδ2− T cells pair with a more diverse array of Vγ segments [34]. This subset is predominant in mucosal tissues, while only a small percentage are present in peripheral blood [25,35,36]. Vδ2− T cells have been shown to exhibit characteristics of memory responses seen in traditional αβ T cells, such as antigen-driven clonal expansions and differentiation from naive to effector phenotypes [26,27,37,38], and they are thus considered adaptive-like in nature. Vδ2− T cells have the ability to recognize tumor cells and virally-infected cells, yet the field is sorely lacking in definitive ligand recognition data for this subset [39].

Both peripheral and tissue resident γδ T cells maintain age-specific functionality and distribution, even in the absence of underlying conditions [40,41,42,43]. γδ T cells are observed as early as 12.5 weeks gestation [44], and there are obvious age-related differences when tracking the frequency and number of γδ T cells across the lifespan. Both frequency and number peak in adulthood, and decrease in aged individuals [40,43,44,45]. Age- and tissue-specific signatures of the repertoire are present as well [46], and the developmental waves of γδ T cells and their TCR repertoires have been recently reviewed [30]. The fetal and infant Vγ9+Vδ2+ repertoires are public, meaning that certain TCR sequences are present across individuals. Non-Vγ9+Vδ2+ repertoires are public during gestation, and become private, or unique to the individual, after birth [42,47]. While the majority of γδ T cells present mid-gestation are Vγ9+Vδ2+, Vδ2− T cells predominate the periphery from late gestation through early infancy [34,41,42,44]. Vδ2+ T cells are present at this time as well, but are not primarily paired with Vγ9+ [40,44]. Vγ9+Vδ2+ T cells begin populating the periphery quickly after birth—the expansion occurs at around 10 weeks of age—and the proportionality seen in adulthood is established by approximately 6 years of age [42,48]. It is believed that the reason for this expansion is two-fold: increased pathogen exposure immediately after birth, and the heightened protective role exhibited by neonatal γδ T cells due to delayed maturation of the αβ T cell compartment [44]. The Vγ9+Vδ2+ expansion during infancy and adolescence is due to proliferation, not increased thymic output [40,48]. Additionally, these cells have repertoires that trace back to fetal γδ T cells based on publicity and a decreased amount of N insertions [34,41,42,44]. While adult γδ T cells show a marked narrowing of the repertoire via decreased segment usage, diversity, and publicity, Vγ9+Vδ2+ T cells maintain publicity in the *TRGV* repertoire, even though the *TRDV* repertoire is private [46,47]. Age-dependent contraction of γδ T cells is limited to the Vγ9+Vδ2+ compartment, while Vδ2− T cells are maintained, if not expanded, upon aging [45,49,50]. Segment usage within the repertoire is also correlated with age and ancestry, as it has been shown that variations in Vδ1+ T cell frequencies are associated with ethnicity (including African, European, and Asian populations) [34,41,45,49,51]. Males have been shown to have more overall γδ T cells than females, and an increase in Vδ2+ T cells is responsible for this difference [49]. The age-dependent distribution and functionality of peripheral γδ T cells is outlined in Figure 1.

When characterizing γδ T cells using activation markers typically studied in αβ T cells [43], it has been reported that virtually all circulating Vγ9+Vδ2+ T cells lose their naive phenotype within the first year of life [40,45,52]. Vδ2− cells at epithelial sites typically maintain a naive phenotype longer, often throughout childhood [40]. The acquisition of cytotoxic markers is variable based on γδ T cell population, location, age, infection status, and even gender and ancestry [38,42,45,49,53]. However, it has been reported that naive markers and activation markers are remarkably comparable between neonates and adults [40]. Cell surface markers alone are not sufficient to accurately assess cytotoxic function, so multiple studies have analyzed the effector functions of γδ T cells at different ages. Interestingly, younger individuals (fetus through 1.5 years of life) have Vγ9+Vδ2+ T cells that are pathogen-reactive in a manner similar to adults, as determined by interferon gamma (IFNγ), granzyme, and perforin production following stimulation [40,44]. While this responsiveness is less robust compared to adult Vγ9+Vδ2+ T cells, any degree of responsiveness in utero, often considered a more sterile environment, is indicative of functional pre-programming that is exposure independent. Vγ9+Vδ2+ T cells expanding through infancy and adolescence have cytotoxic effects that are comparable to adults, such as the expression of both Perforin and Granzyme B [42]. Upon aging, the responsiveness of this subset to stimulation deteriorates. Aging Vγ9+Vδ2+ T cells express upregulated CD69, and have higher basal expression of tumor necrosis factor alpha (TNFα) compared to younger individuals. However, these cells show a decreased responsiveness to IPP in culture based on TNFα production [54]. These results reveal that γδ T cells are not immune to the effects of immunosenescence and inflamm-aging. Disease history is another driver of γδ T cell functionality, and in the following section we will outline the effect of age on disease-responsive γδ T cells. For a more detailed description of γδ T cell effector functions in both health and disease with reference to age, see Table 1. It is important to note that effector functionality of γδ T cells has not been thoroughly investigated with respect to age.

## 3. Reactive γδ T Cell Immunity

### 3.1. Cancer

The role of human γδ T cells in cancer is widely variable based on the type and location. γδ T cells are able to infiltrate the tumor microenvironment, as observed in melanoma [77], rectal cancer [80], and breast cancer [81]. The anti-tumor effects of Vγ9+Vδ2+ T cells are present early in life, as a study in 1990 revealed that cord blood Vγ9+Vδ2+ T cells were responsive to a Burkitt’s lymphoma cell line [82]. Both Vγ9+Vδ2+ and Vδ1+ T cells exhibit anti-tumor effects in both TCR-dependent and NK cell receptor (NKR)-dependent mechanisms, with downstream effects including perforin and granzyme expression, tumor necrosis factor-related apoptosis-inducing ligand (TRAIL) expression, human apoptosis-related factor ligand (FASL)-mediated apoptosis, and antibody-dependent cell-mediated cytotoxicity (ADCC) [31]. The majority of studies concerning the efficacy of γδ T cells in cancer treatment focus on the Vγ9+Vδ2+ T cell subset [83,84,85,86,87], due to the incomplete understanding of Vδ1+ T cell recognition. While some Vδ1+ γδ T cells can promote an anti-tumor environment as mentioned above, others appear to promote a pro-tumor environment [88]. IL-17A-producing γδ T cells (Tγδ17 cells) are the primary pro-tumor γδ T cell subset. These cells are typically Vδ2−, although a small percentage of Vγ9+Vδ2+ T cells have been shown to express IL-17A [56,58]. Broadly speaking, Tγδ17 cells exhibit immunosuppressive functions and promote tumor growth. This immunosuppression can target T cells and dendritic cells, allowing tumor cells to escape immune surveillance [80,88,89]. IL-17A production can dismantle the anti-tumor functionality of IFNγ-producing γδ T cells, and Tγδ17 cells have been associated with tumor progression and poor outcomes in multiple tumor types [31,88]; however, the mechanism by which this occurs in humans is not fully understood. In addition to IL-17A production, IL-4 expression has been shown to inhibit the anti-tumor response of Vγ9+Vδ2+ T cells via the promotion of IL-10-producing Vδ1+ T cell growth [59]. It is important to note that these rules are not hard and fast, as a small number of mouse studies have implicated Tγδ17 cells in a tumor-protective role [88].

On their own, none of the aforementioned data point to an age-related association between γδ T cells and cancer immunity. Using the knowledge that Vδ1+ T cells are predominant in the periphery of infants and Vγ9+Vδ2+ T cells largely constitute the periphery of adults, one could speculate that adulthood promotes an anti-tumor functionality of γδ T cells. Since the γδ T cell contraction in the elderly is restricted to Vγ9+Vδ2+ T cells while the Vδ2− compartment remains intact, one could additionally speculate that aging promotes a pro-tumor environment. The same could be said for both tissue-resident and mucosal γδ T cells, as these locations are home to Vδ1+ T cells and IL-17A-producing γδ T cells, respectively. Additionally, aging results in increased inflammation, and it is likely that Tγδ17 cells elicit their pro-tumor functions in an inflammation-dependent manner [88]. However, this inference is muddied by the reality that both subsets have been shown to exhibit pro- and anti-tumor immunity, and we were not able to find a study that has reported the impact of age on the γδ T cell response to cancer.

### 3.2. Transplant

γδ T cells have been studied in the context of transplant since the 1990s, and are an important subset to consider in the field due to TCR-dependent tissue localization and a lack of MHC restriction [90]. They have been shown to quickly reconstitute along with NK cells following allogeneic hematopoietic stem cell transplant (HSCT), and are believed to effectively fill the role of αβ T cells, which take longer to recover [91,92,93,94]. γδ T cells, in the context of transplant, have been reviewed more thoroughly elsewhere [73,90,95], revealing their multifaceted effect on overall success and survival. Following HSCT, positive correlations have been reported between elevated γδ T cell numbers and increased survival rate, while a negative correlation has been reported between γδ T cell numbers and graft-versus-host-disease (GVHD) severity. When immune ablation for transplant results in primary human cytomegalovirus (HCMV) infection or reactivation, γδ T cells are typically associated with favorable outcomes. Despite these data, γδ T cells have also been correlated with negative transplant outcomes, as studies in the 1990s revealed that Vδ2+ T cells were enriched in patients with acute GVHD three months post-transplant [73]. It has been reported that Vγ9+Vδ2+ T cells are reconstituted with a highly similar repertoire following HSCT, while Vδ1+ T cells exhibit a skewed repertoire [96]. The results of a pediatric HSCT study in 2015 aligned with reported adult HSCT data, where increased γδ T cell reconstitution was significantly correlated with both increased event-free survival and decreased post-HSCT infections [92]. While we were unable to locate a study outlining the unique features distinguishing pediatric and adult γδ T cell responses following transplant, Witte et al. reported that the transplant graft donor age can affect γδ T cell reconstitution. For example, Vδ2+ T cells are virtually absent in patients receiving umbilical cord grafts [94]. Based on the age- and infection-dependent distribution of γδ T cell subsets, we hypothesize that donor age and herpesvirus serostatus would greatly influence the repertoire of reconstituted γδ T cells, shaping the immune system for years to come.

### 3.3. Infectious Disease

Numerous studies have highlighted the importance of γδ T cells in infectious disease. Their function is determined by factors including route of infection and tissue dissemination. Comprehensive overviews of γδ T cells in viral and bacterial infections have been published over the years [64,97,98,99], yet there is minimal mention of the role that age plays in these responses. In this section, we intend to outline the effects of age on the human γδ T cell response to pathogens that have been reported.

#### 3.3.1. Cytomegalovirus

Human cytomegalovirus (HCMV) is a latent herpesvirus with immunomodulatory capabilities across the lifespan [100,101,102]. This virus has an infectivity rate of 30–90% [103], and it is estimated that one in three children are infected by the age of 5 [104]. The age-dependent immune response to HCMV is documented in multiple cell types, including CD8+ T cells and NK cells [102,105,106,107,108,109,110]. It is well-established that γδ T cells also respond to HCMV, whether in a natural infection/latency/reactivation cycle [103,111,112], or in response to the immune ablation of transplant patients [73,90,113,114]. While Vγ9+Vδ2+ T cells are typically the most prominent γδ T cell subset in the periphery, CMV has been shown to shift that axis. This is due to the expansion and response of Vδ2− T cells (typically Vδ1+ and Vδ3+) [74,115,116].

The effect of age on γδ T cell immunity to HCMV is arguably the most well-characterized [103,114], and is outlined in Figure 1. When HCMV infection occurs in utero, an expansion, activation, and CDR3 restriction of Vγ9− T cells is seen, including Vδ1+, Vδ2+ and Vδ3+ T cells [103,114,117]. A public Vγ8+Vδ1+ clone has also been identified in the fetal response to HCMV as early as 21 weeks of gestation [117]. In adults, HCMV has no observable effect on Vδ2+ T cells [103], and the Vγ8Vδ1 T cell compartment is unaffected by HCMV serostatus [114]. Importantly, even the public fetal Vγ8+Vδ1+ clone is absent in the HCMV+ adults [117]. The most striking effect of HCMV on γδ T cells in aging individuals is within the Vδ2− T cell compartment, even after the clearance of lytic infection [38,114]. As mentioned previously, aging results in a decrease in γδ T cell frequency and number. In the absence of HCMV, this decrease is observed in both Vδ2+ and Vδ2− compartments [74,115], although some studies report that the age-dependent contraction of γδ T cells is limited to the Vγ9+Vδ2+ T cell compartment alone [45,49,50,112]. In the presence of HCMV, the Vδ2+ T cell compartment shrinks as expected, but the Vδ2− T cell compartment remains largely unchanged, if not expanded [103,112,113,115]. Additionally, it has been reported that CMV seropositive individuals maintain a consistent Vδ1+ population that is not affected by age to the same degree as the same compartment in seronegative individuals [115,116]. In response to HSCT, Vδ1+ T cells robustly respond to CMV reactivation, in a manner that is not observed in CMV− individuals, or in CMV− EBV+ patients [114]. A study published in 2013 argued that the age-dependent shaping of γδ T cells in elderly individuals is almost exclusively driven by HCMV serostatus [116].

The HCMV-driven mobilization of γδ T cells occurs in both immunocompetent and immunocompromised individuals [38]. To date, studies analyzing the γδ T cell response to active HCMV in vivo are all performed in a transplant setting, studying reactivation. This is because it is virtually impossible to determine when primary HCMV infection occurs in healthy individuals. However, studying both activation status and cytotoxic effects of seropositive individuals in vitro reveals a long-term signature on γδ T cells. Vδ1+ T cells have been shown to both kill HCMV-infected cells and limit viral propagation via the expression of TNFα [103,116,118]. HCMV-activated Vδ2− T cells also express IFNγ [27,40,103], and the virus drives Vδ2− T cells from consisting of both naive and effector memory cells to being predominantly effector memory [74]. Fetal γδ T cells responding to HCMV were also able to express IFNγ, and they differentiated into an activated status comparable with CD8+ T cells [117]. Candidate ligands for Vδ1+ T cells have been identified, and they are diverse in nature [26]. One in particular, endothelial protein C receptor (EPCR), expressed on HCMV-infected cells, has been directly recognized by a Vγ4+Vδ5+ clone [39,119]. While we could not find any data on the role of age and ligand recognition in CMV infection, the data outlined in this section clearly indicate an interconnectedness between HCMV, γδ T cells, and age.

#### 3.3.2. Influenza

It has been well-established that age has a dramatic effect on the immune response to influenza [120,121,122]. This immunomodulation is seen in both the ability to clear the virus, as well as vaccination efficacy [123,124]. The γδ T cell response to influenza is vastly different when compared to the HCMV response. Where Vδ2− T cells are the main HCMV responders, Vγ9+Vδ2+ T cells are the primary responders in both human and avian influenza virus infection [46,65,125,126]. The majority of studies into the γδ T cell effector response to influenza are performed in vitro, using different forms of stimulation. Vγ9+Vδ2+ T cells cultured with pAgs that are both endogenous [125,126] and exogenous [127,128], as well as cells incubated with virus, have potent cytotoxic effects against influenza-infected cells. This is primarily accomplished via IFNγ production by the stimulated Vγ9+Vδ2+ T cells. pAg stimulation does seem to increase the potency of influenza-reactive Vγ9+Vδ2+ T cells, and it has been proposed to use pAg treatment in patients as a means to increase γδ T cell reactivity [65,127]. All of the aforementioned studies utilized γδ T cells isolated from healthy donors, presumably adults, so no correlations with age can be made. A recent study in 2019 compared the adult γδTCR repertoire between healthy and influenza-infected adults, revealing that influenza-reactive Vγ9+Vδ2+ T cells exhibit an enrichment of public Vγ9+ clonotypes with IFNγ-production capabilities [46]. Interestingly, the comparison also revealed a public Vγ9+ CDR3 sequence in influenza-responsive cells that was present in all healthy cord blood samples and adult donors, but not in every healthy elderly donor [46]. Peripheral γδ T cells are also able to respond to influenza vaccination in an age-influenced manner, as younger adult populations exhibit a more robust response when compared to elderly individuals [129]. That said, the human γδ T cell field is still vastly underrepresented in studies concerning age and its role in influenza responses.

In CD8+ T cell biology, the presence of CMV can positively affect the immune response to influenza in both mice and humans [124]. This effect was age-specific, as increased age led to a decreased responsiveness in CD8+ T cells, even in the presence of CMV. Age is an important factor in understanding how co-infection shapes the immune system [130,131], yet γδ T cells are sorely underrepresented in co-infection studies. One could speculate that there may be a similar effect of CMV on influenza-responsive γδ T cells; however, these two viruses elicit responses from different γδ T cell subsets. Additionally, age has a unique influence on each of the two γδ T cell compartments in question.

#### 3.3.3. Tuberculosis

The effects of both *Mycobacterium tuberculosis* (*Mtb*) infection and Bacillus Calmette–Guérin (BCG) vaccination on γδ T cells have been studied since the late 1900s [42,132,133,134], and Vγ9+Vδ2+ T cells are the primary subset of γδ T cells that respond to tuberculosis (TB) infection [75,135]. TB has been shown to shape the Vγ9+Vδ2+ TCR repertoire, as TB patients have a CDR3δ that is more polyclonal than healthy donors [136]. Additionally, novel CDR3 sequences were predominant in TB patients alone [136]. A public Vδ2+ CDR3 has been reported across adult TB patients [76], and potential ligands have been identified as well [135,136,137]. BCG vaccination has also been shown to elicit a robust Vγ9+Vδ2+ T cell response in adults [138,139], and challenges using TB antigens in vitro resulted in a memory-like expansion of IFNγ-producing Vγ9+Vδ2+ T cells [37]. This response seems to be age-specific, because vaccination early in life does not affect the expansion and activation of neonatal Vγ9+Vδ2+ T cells [42], although Vδ2+ T cells from cord blood are able to respond in vitro to both exogenous BCG and heat-inactivated *Mtb* [140,141]. Generally, age has been found to be a factor in the response to TB [142,143], especially in the elderly due to increased basal inflammation [144]; however, the role of γδ T cells in this age-dependent response is not fully elucidated.

#### 3.3.4. Miscellaneous Microbes

The γδ T cell response to bacteria is innate-like in nature, and also involves Vγ9+Vδ2+ T cells [67,99]. This compartment responds to both foreign (HMB-PP) and self (IPP) pAgs [97,145]. It is believed that the microbe-driven response of γδ T cells is germline encoded, as enrichment of pAg-reactive Vγ9+Vδ2+ T cells is present in the fetus during the second trimester [44]. This responsiveness is comparable to what is seen immediately after birth [146], indicating that microbial exposure is not necessary to elicit a response. These fetal cells express a public germline sequence seen in 50% of Vγ9+Vδ2+ T cells, and they are able to rapidly respond to pAg stimulation [44]. As was true with influenza-responsive Vγ9+Vδ2+ T cells, the majority of human studies concerning the γδ T cell responsiveness to microbes is accomplished in vitro, typically utilizing endogenous and/or exogenous pAgs. Because of this, the field is sorely lacking in age-specific data, especially when considering how microbial infections shape the γδTCR repertoire and effector functions across the lifespan.

#### 3.3.5. Miscellaneous Chronic Infections

Chronic infections uniquely shape the immune system, and the interplay between age, coinfection, and disease is dynamic in nature. Not all chronic viral infections are created equally, and can be loosely sorted into three categories: latent, smoldering, and persistent [131]. γδ T cells have been implicated in the immune response to chronic viruses in each of these categories, and their functionality is contextual. For example, Vγ9+Vδ2+ T cells retain the ability to both kill and inhibit replication of cells infected with latent viruses like Epstein–Barr virus (EBV) and herpes simplex virus (HSV) [64,99,147]. As outlined above, HCMV is a smoldering chronic infection in which γδ T cells have a lasting effect that is age-dependent. γδ T cells have also been implicated in the immune response to human immunodeficiency virus (HIV), hepatitis B virus (HBV), and hepatitis C virus (HCV), all of which are persistent chronic viruses [71,79,147,148]. However, the functionality of γδ T cells in each of these instances is unique, and in some cases even conflicting [149]. With the exception of murine cytomegalovirus (MCMV), there is limited data reporting the effects of age on chronic viral immunity. One such study, which characterized γδ T cell inhibitory receptor signatures in HIV infection, identified a link between elevated inflammation and aviremic HIV that is exacerbated in aging individuals [150]. There is a general paucity of human studies investigating the γδ T cell response to chronic viruses, making it difficult to speculate about a pattern of γδ T cell effector functionality across chronic infections based on segment usage, but this should be an area for further research in γδ T cell biology.

## 4. Discussion

γδ T cells have been implicated in a multitude of other disease states, including the viruses West Nile and SARS-CoV-2 [151,152,153], parasitic infections including *Plasmodium falciparum* [154], and non-infectious diseases such as multiple sclerosis and arthritis [155,156,157]. Since no age-dependent γδ T cell responses are reported for these disease states, they fall outside the scope of this review. While models of immune functionality and aging in rodents are much more well-established, humans were the primary focus of this review because there is not a subset of γδ T cells in mice that are pAg-reactive and comparable to human Vγ9+Vδ2+ T cells [148]. A more complete understanding of the relationship between age and γδ T cell functionality in health and disease requires a deeper inter-species analysis that includes both γδ T cell-low (humans, mice) and γδ T cell-high (cattle, sheep, chickens, rabbits) species [16]. Until then, we have outlined clear age-dependent responses in γδ T cells, both in passive and reactive immunity. Additionally, we have highlighted gaps in the field that clearly indicate there is much left to explore concerning how age affects γδ T cells. To fully understand the complexity of the immune system across the lifespan, a clear understanding of this cell type is necessary.

## Figures and Tables

**Figure 1 ijms-21-08903-f001:**
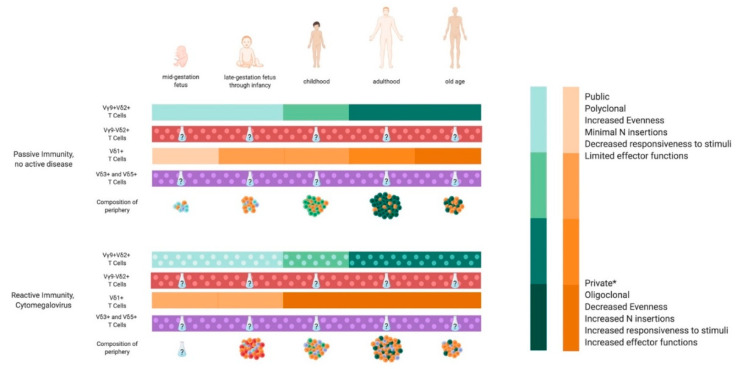
An overview of the effects of age and cytomegalovirus (CMV) on the peripheral γδ T cell compartment in humans, based on segment usage. Due to the lack of data concerning age and segment usage over time in the context of most diseases, human cytomegalovirus (HCMV) is the only disease state reported in the figure. Solid colors are indicative of reported data, while patterned colors signify a lack of information. This lack of information stems from either insufficient data on the specified subset, or that the effects of HCMV on the population have not been characterized because the subset in question has not been found to be directly involved in the anti-HCMV immune response. Created using BioRender.com * Vγ9+Vδ2+ T cells become increasingly private with age, but public *TRGV* sequences persist into adulthood and beyond while *TRDV* sequences become fully private.

**Table 1 ijms-21-08903-t001:** Overview of reported γδ T cell effector molecules based on age and location in health and disease.

Subset(s)	Location	Age	Effector Molecules	Disease State *
Vδ2+	Periphery	Cord Blood	IFNγ **	Homeostasis [55]
Vδ2+	Periphery	Neonates (14 d)	IFNγ	Homeostasis [55]
Vδ1+Vδ2+	Periphery	Infant (4 m)	IFNγ, perforin	Homeostasis [53]
Vγ9+Vδ2+	Periphery	Infant (1 y)	IFNγ	Homeostasis [53]
Vδ2+	Periphery	Children (2–5 y)	IFNγ	Homeostasis [55]
Vγ9+Vδ2+	Periphery, Cerebrospinal fluid	Children (3–14 y)	IL-17A **	Bacterial meningitis [56]
Vγ9+Vδ2+	Periphery, Tonsils	Children (9–14 y)	IL-2, IL-4, IL-10	Tonsillectomy patients [57] (Caccamo 2006)
Vγ9+Vδ2+	Periphery	Adult	IFNγ, TNFα **, IL-2, IL-4, IL-10, IL-17A	Homeostasis [26,53,56,57,58,59,60,61]
Vδ1+	Periphery	Adult	Perforin, granzymes	Homeostasis [26]
Unspecified	Periphery	Adult	IFNγ, TNFα, IL-6, IL-17A, IL-10	Homeostasis [59]
Unspecified	Mucous Membrane	Adult	IL-10, TGF-β **, TNFα, IFNγ, IL-4, IL-2	Pregnancy [62]
Vγ9+Vδ2+	Periphery	Adult	IFNγ, TNFα, IL-17A	Rheumatic disease [63]
Vγ9+Vδ2+	Periphery	Adult	IFNγ, CCL3 **, CCL4, CCL5	Influenza [64,65,66]
Vγ9+Vδ2+	Periphery	Adult	IFNγ, TNFα	Legionellosis [67]
Vγ9+Vδ2+	Periphery	Adult	IFNγ	Human immunodeficiency virus (HIV) [68]
Vγ9+Vδ2+	Lesions	Adult	IL-17A, IL-8, IFNγ, TNFα	Psoriasis [69]
Vγ9+Vδ2+	Tumor-infiltrating	Adult	IFNγ, TNFα	Colon carcinoma [70]
Vδ2+	Periphery	Adult	IFNγ, IL-17A	Hepatitis B virus (HBV) [71,72]
Vδ2−	Periphery	Adult	IFNγ, TNFα	Human cytomegalovirus (HCMV) [64,73,74]
Vδ1+	Periphery, Synovial fluid	Adult	IFNγ, IL-4	Rheumatic disease [63]
Unspecified	Periphery	Adult	IFNγ, TNFα, IL-4, IL-10	*Mycobacterium tuberculosis* [75,76]
Unspecified	Periphery	Adult	IL-17A, TNFα	Various bacterial infections, *Plasmodium falciparum* [64]
Unspecified	Tumor-infiltrating	Adult	IFNγ, TNFα, IL-17A, IL-4, TNFβ	Miscellaneous adult cancers [77,78]
Unspecified	Liver-derived celllines	Adult	IFNγ, TNFα, IL-8	Viral hepatitis [79]

* Many of the reported effector functions are derived from in vitro treatments (e.g., stimulation with virus, phorbol myristate acetate (PMA)/Ionomycin, phosphoantigen (pAg), etc.). Some studies were done using primary cell lines. ** IFN = interferon, IL = interleukin, TNF = tumor necrosis factor, CCL = C-C motif chemokine ligand

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
