# Peer review of "A Cell for the Ages: Human γδ T Cells across the Lifespan"

_ijms, 2020, doi:10.3390/ijms21238903_

Round 1

Reviewer 1 Report

In my opinion the presented paper “A cell for the ages: human γδ T cells across the lifespan“ submitted by Brandi Clark and Paul Thomas fits into the scope of the International Journal of Molecular Sciences.
In general, the content and the form of the presented data are fine. The scientific community will benefit from this summarizing picture of the versatile nature of human γδ T cells.
However, the manuscript would benefit from adding comparative aspects with other species than rodants, e.g. companion animals and livestock such as pig, ruminants or poltry.
It might be worth to discuss the immunological significance of γδ T cell low versus γδ T cell high species, primarily in the context of infectious diseases.

Author Response

Point 1: However, the manuscript would benefit from adding comparative aspects with other species than rodents, e.g. companion animals and livestock such as pig, ruminants or poultry.
It might be worth to discuss the immunological significance of γδ T cell low versus γδ T cell high species, primarily in the context of infectious diseases.

Response 1: Thank you for this helpful comment. The concept of comparative γδ T cell biology between species is intriguing to us, and a thorough review of the  comparative elements of γδ T cell recognition would be of benefit to the field. However, the focus of our review is on γδ T cell biology and variation across the age span, and this aspect in many other species (mice excluded) is poorly covered in the literature. There does not seem to be any published work concerning age in rudimentary animals, for example. As a result, we have added discussion about broad comparisons between species and the need for additional work in this area on lines 357 to 362.

Reviewer 2 Report

The review by Clark et al describes the biology of human γδ T cells over lifetime in various disease conditions. This review depicts the important biological aspects of γδ T cells, however there is a need to explain some parts in more depth for better understanding. Please see my specific comments below.

  1. The subsets of γδ T cells are complex and it would be appropriate to represent different subsets, their effector/cytotoxic molecules and distribution in different tissues in the form of a table or figure.
  2. The development of γδ T cells according to age is described in line 93-120 and in Fig. 1. This information read complicated in the current form. It is also hard to understand the Fig. 1. It would be better to elaborate the current information and simplify the Fig.1 by representing cell frequencies with age in a graph form.
  3. Line 112-113: please expand TRDV and TRGV.
  4. The frequently used terms ‘private’ and ‘public’ for TCR repertoires should be explained.
  5. It would be interesting to know the functions of γδ T cells during human chronic viral infections such HIV, HBV or HCV and its relation with aging.

Author Response

Point 1: The subsets of γδ T cells are complex and it would be appropriate to represent different subsets, their effector/cytotoxic molecules and distribution in different tissues in the form of a table or figure.

Response 1: We thank the reviewer for this suggestion and have now added a Table summarizing key features of different subsets.

Point 2: The development of γδ T cells according to age is described in line 93-120 and in Fig. 1. This information read complicated in the current form. It is also hard to understand the Fig. 1. It would be better to elaborate the current information and simplify the Fig.1 by representing cell frequencies with age in a graph form.

Response 2: Thank you for this feedback. We admit that this information is difficult to outline in a simplified form, and we have made minor changes to this section—both in the body of the paper and the figure legend (see lines 95 to 122 and 147 to 155). In reference to your idea concerning the graph, we were intrigued by this suggestion but were unable to accurately summarize the data in this way. While advances in γδ T cell biology are constantly occurring, there are still a myriad of gaps in the field. The robust quantification of cell frequencies across the age span necessary to generate accurate graphs are not present in the literature we reviewed. The current literature outlined in our graph is representative of reported trends, but variability within these trends themselves coupled with inherent variability in human research result in an inability to procure specific cell frequencies across segment usage and age. Additionally, segment usage remains an important portion of the figure to the authors, as the reported abundance of each group of γδ T cells differs based on age.

Point 3: Line 112-113: please expand TRDV and TRGV.

Response 3: Thank you for pointing out this oversight. TRDV and TRGV have been appropriately expanded, and can be found on lines 77 and 78.

Point 4: The frequently used terms ‘private’ and ‘public’ for TCR repertoires should be explained.

Response 4: Thank you for pointing out that public and private were not explicitly explained. A definition for each term can be found on lines 101 to 103.

Point 5: It would be interesting to know the functions of γδ T cells during human chronic viral infections such HIV, HBV or HCV and its relation with aging.

Response 5: Thank you for this comment, and we certainly agree! Unfortunately, there is a paucity of data concerning the relationship between age and chronic viruses, with the exception of HCMV. That said, we agree that a discussion of chronic viral infections and γδ T cell responses across the age spectrum is warranted and can be found under sub-heading 3.3.5 on lines 334 through 351.

Round 2

Reviewer 2 Report

My all concerns have been addressed by the authors.